# Central Nervous System Pericytes Contribute to Health and Disease

**DOI:** 10.3390/cells11101707

**Published:** 2022-05-20

**Authors:** Francesco Girolamo, Mariella Errede, Antonella Bizzoca, Daniela Virgintino, Domenico Ribatti

**Affiliations:** 1Unit of Human Anatomy and Histology, Department of Basic Medical Sciences, Neuroscience and Sense Organs, University of Bari ‘Aldo Moro’, 70124 Bari, Italy; mariella.errede@uniba.it (M.E.); daniela.virgintino@uniba.it (D.V.); domenico.ribatti@uniba.it (D.R.); 2Physiology Unit, Department of Basic Medical Sciences, Neuroscience and Sense Organs, University of Bari ‘Aldo Moro’, 70124 Bari, Italy; antonella.bizzoca@uniba.it

**Keywords:** Alzheimer’s disease, angiogenesis, mesoangioblast, neurodevelopmental disorders, neuroinflammation, neurovascular unit, multiple sclerosis, neuroCOVID-19, stroke

## Abstract

Successful neuroprotection is only possible with contemporary microvascular protection. The prevention of disease-induced vascular modifications that accelerate brain damage remains largely elusive. An improved understanding of pericyte (PC) signalling could provide important insight into the function of the neurovascular unit (NVU), and into the injury-provoked responses that modify cell–cell interactions and crosstalk. Due to sharing the same basement membrane with endothelial cells, PCs have a crucial role in the control of endothelial, astrocyte, and oligodendrocyte precursor functions and hence blood–brain barrier stability. Both cerebrovascular and neurodegenerative diseases impair oxygen delivery and functionally impair the NVU. In this review, the role of PCs in central nervous system health and disease is discussed, considering their origin, multipotency, functions and also dysfunction, focusing on new possible avenues to modulate neuroprotection. Dysfunctional PC signalling could also be considered as a potential biomarker of NVU pathology, allowing us to individualize therapeutic interventions, monitor responses, or predict outcomes.

## 1. Introduction

Pericytes (PCs) are mural cells that wrap, support, and control blood microvessels [1]. There is evidence that they are also present in arterial and venous walls [2,3,4]. PCs can be distinguished from other perivascular cells by a combination of criteria, including their anatomical location embedded within the vascular basement membrane, and their morphology and gene/protein expression pattern [5,6]. At the ultrastructural level, PCs establish direct contact with endothelial cells (ECs) [5,6,7], and the basement membrane (BM) is often absent in places where PCs and ECs are in juxtaposition [8]. The BM shared between PCs and ECs is co-produced, and composed primarily of collagen type-IV, -VI and laminins [9]. During development, the newly formed capillaries attract PCs secreting the chemoattractant platelet-derived growth factor (PDGF)-BB, which binds to the PDGF receptor β (PDGFRβ) expressed on PCs [10,11]. Integrin-mediated adhesion of PCs to laminin is necessary to maintain a constant expression of PDGFRβ [5,12,13]. Interference with this PDGF-BB/PDGFRβ signalling is sufficient to disrupt EC–PC interactions [5,14], indicating that perpetual signalling through PDGFRβ is crucial to maintain the PC localization on the abluminal surface of microvessels. Several other proteins, such as Notch receptors, support the continual close interaction of PCs with the underlying ECs, and conserve the PC identity [15].

## 2. PC Identity, Multipotency and Differentiation Potential

PCs are a heterogeneous cell population encompassing different subtypes that exert different functions. Firstly, PCs have been classified into three subtypes: precapillary, midcapillary and postcapillary, based on their location and morphology [16]. Precapillary PCs bear several circular branches wrapping around blood vessels. Midcapillary PCs are long, spindle-shaped cells, which mainly extend parallel to the length of microvessels and have many short secondary processes. Postcapillary PCs are short stellate-shaped cells that cover the abluminal surface of postcapillaries and postcapillary venules [17] (Figure 1). The PC subtypes feature different expressions of alpha smooth muscle actin (α-SMA): midcapillary PCs show a lower α-SMA expression than pre- and postcapillary PCs [17,18].

An additional classification of skeletal muscle PCs identifies two alternative subtypes, type 1 and type 2 PCs, depending on their different expression of Nestin or PDGFRα [20,21,22]. Another important feature of PCs is their degree of maturity or stemness. PCs residing in a stable microvessel network are primarily quiescent. However, physiological expansion of the vasculature in conditions of increasing metabolic requirements, and a variety of pathological conditions, trigger PC activation and proliferation, leading to self-renewal or differentiation. Maintenance of the self-renewal and multipotent state of PCs is highly dependent on interactions with the BM protein laminin. In the absence of this protein, brain PCs develop the properties of hyper-differentiated PCs, α-SMA^+^ contractile cells enwrapping the central nervous system (CNS) microvasculature [23]. On the contrary, when specifically cultured, PCs can differentiate into multiple lineages [24,25]. This PC multipotency resembles the capacity of “mesenchymal progenitors” or “mesoangioblasts”. In fact, cultured PCs display an even broader multipotency than mesenchymal progenitors; they migrate by BM digestion [24,25,26,27,28,29,30]] and are able to differentiate into vascular smooth muscle cells (SMCs) [31,32], osteoblasts [33], chondrocytes [34], adipocytes [22,25], macrophages [35,36], myofibroblasts [22,29,30], myoblasts [37], myocardiocytes [27] and neuronal and glial cells comprising oligodendrocyte precursor cells (OPCs) [38,39,40,41,42,43]. The hypothesis that PCs are the predominant population of tissue-resident mesenchymal progenitors derives from their broad multipotency [24,44]. This multipotency has also led to their use in cell transplantation in regenerative medicine [45,46], although this concept has recently been challenged by studies of T-Box transcription factor 18 (Tbx18)^+^ mouse, in which PCs maintain their perivascular identity without generating any other cell types in injured tissue [28,47]. Accordingly, multipotent PCs are marked by the absence of Tbx18 [28,48]. In addition, specific PC subtypes within tissues exhibit distinct transcriptomes and differentiation potentials, that may correspond to a pre-programmed commitment to specific lineages [38,46,49]. Single cell studies profiling of brain- and lung-derived PCs support this view [50,51]. The expression profile of some PC regulators appears to be lineage-specific, such as Runt-related transcription factor 2 (Runx2) for osteogenesis, peroxisome proliferator-activated receptor γ (Ppar-γ) for adipogenesis, and SRY-box transcription factor 9 (Sox-9) for chondrogenesis [9]. Furthermore, single-cell sequencing identified sub-populations of adult brain-derived PCs that can reprogram neuronal differentiation [52]. Brain PCs are a potential source of precursors that can regenerate neuronal cells [53,54]. However, this great differentiation potential varies according to the specific tissue/organ of origin [49,55,56]. For example, PDGFRβ^+^ zinc finger protein 423 (ZFP423)^+^ PCs of murine adipose tissue readily differentiate into adipocytes, thus contributing to adipocyte hyperplasia [57]. PC differentiation is epigenetically regulated by patterns of tissue-specific histone modification pattern within genes that are known to regulate the PC phenotype, metabolism and fate [49]. These data indicate that PC subsets have a certain degree of pre-programmed commitment to specific lineages, although PC multilineage differentiation has been demonstrated only in vitro. It is still a matter of debate whether PCs in vivo also receive sufficient microenvironmental signals to promote these various differentiation fates in physiological or pathological conditions [48]. The mechanisms regulating this multipotency and tissue-specific pre-programming require further elucidation.

## 3. PC Markers

To date, there is no unique molecular marker specific for PCs. Thus, a combination of criteria is used to define PC populations, such as a perivascular localization in microvessels, the morphology and the expression of one or more recognized molecular markers such as Neural/glial antigen 2 (NG2), PDGFRβ or cluster of differentiation 146 (CD146) [5,58,59]. However, these PC markers lack unequivocal specificity. They are expressed to some extent in other cell types (i.e., OPCs, SMCs and fibro-adipocyte progenitors) and they also display variable expression patterns on PCs across different tissues, as well as different locations within the vascular tree, developmental state and pathological setting (Table 1) [16,46,60]. In recent years, remarkable progress has facilitated the identification and characterization of PCs in several tissues, using the combination of confocal microscopy with genetic state-of-the-art techniques, but some unanswered questions remain about their origin and functions in health and disease [61]. PCs probably share a common mural precursor with the vascular SMCs of the specific tissue in which they both reside [5,12]. Notably, the identification of PCs may be complicated by their heterogeneity, which depends not only on their different origin and distribution but also on their organ-specific pattern and dynamic expression of molecular markers [5,62].

## 4. PC Origin

Although PCs were firstly observed and described more than 150 years ago [77,78,79], multiple lineage-tracing experiments on PC origin during embryogenesis have shown that the cellular sources of PCs are heterogeneous in different tissues [5,12,13]. PCs in the CNS and thymus develop from the neural crest [5,14,80,81,82,83,84], while PCs in the heart, lung, liver and gut originate from the mesothelium [85,86,87]. Mesothelial cells in coelomic organs undergo the epithelial-to-mesenchymal transition, delaminate and migrate into the organs to produce their mesenchymal components, including fibroblasts, vascular SMCs and PCs [15]. Mesodermal mural cells originate from a clonal precursor mesoangioblast through differentiation into a primordial PDGFRβ^+^/CD271^+^/CD73^−^ mesenchymal progenitor [16]. These mesodermal-derived PCs can be further distinguished into CD274^+^ capillary or Delta Like Non-Canonical Notch Ligand 1 (DLK1)^+^ arteriolar PCs, with a proinflammatory or contractile phenotype, respectively [16].

CNS PCs are derived primarily from the neural crest [62] and comprise the population destined to constitute the forebrain, while PCs of mesodermal origin may be found in the brainstem and spinal cord [81,88]. The neuroectodermal origin of PCs was confirmed using Wingless-Type MMTV Integration Site Family, Member 1 (Wnt-1)-Cre recombinase [89] and Sry-related HMg-Box gene 10 (Sox10)-Cre [90] fate-mapping mouse models. Other CNS PCs originate from the bone marrow [20,62,91], macrophage lineage [92]; bone marrow-derived PCs also contribute to the corneal vasculature [93] and to the liver health and disease [94]. Furthermore, regardless of their origin, both mesoderm-derived somite cells and neural crest-derived frontonasal mesenchymal cells are switched to a perivascular cell fate by constitutive Notch-1 activation [95]. In the wall of the aorta, vascular SMCs have multiple developmental origins, including from the neural crest, somite and secondary heart field [5,12,13].

## 5. PC Functions

Regardless of their tissue of origin, PCs regulate blood vessel flow and permeability, vascular development, maturation, regeneration and stabilization [5,10,96,97,98,99,100,101,102]. PCs are generally required to maintain the stability and maintenance of baseline vessels in adults, although, surprisingly, selective PC loss in stable adult retinal vessels does not cause blood-retinal barrier disintegration, indicating that they have an organ and context-specific role in vessel stabilization [103]. The close interaction between PCs and ECs promotes the integrity and quiescence of ECs through their physical connections and via the secretion of paracrine factors, i.e., angiopoietin-1, tissue inhibitor of metalloproteinase-3 (Ang-1, TIMP-3) [104,105]. Conversely, loss of PC contact generally reduces EC survival and promotes capillary regression [45,46]. PCs have additional functions in the vascular compartment, including the preservation of capillary barrier function, blood flow regulation and immunomodulation [5,62]. PCs also display several other different tissue-dependent functions and contribute to other cellular processes involved in tissue homeostasis through their potential to differentiate into other cell types, as already reported.

## 6. CNS Microvascular PCs and Potential Biomarkers of NVU Damage

An attractive growing field of PC biology is the study of their functions and dysfunction in the CNS, also in view of the that they are targeted in the brain and spinal cord diseases. PCs have emerged as a central component of the so-called neurovascular unit (NVU), a platform for safe communications of neurons, glial cells, PCs and ECs. A definite separation of the neuropil from most molecules circulating in the blood stream is critical for neuronal homeostasis and brain functions. The blood–brain barrier (BBB) performs this protective role. The establishment and integrity of a functional BBB during CNS development and adulthood are assured by multiple interactions among the different NVU components: peg and socket connections through N-cadherin, heterotypic gap junctions (GJs), the BM shared by PCs and ECs, the extracellular matrix (ECM) molecules and soluble factors interplay between PCs and ECs, the presence and influences of perivascular astrocytes, microglia, OPCs, macrophages and nerve terminals [62,69,106,107,108,109,110,111,112,113,114,115,116]. CNS PCs express all PC markers of the other tissues and organs, together with a few other specific features such as the expression of the potassium inwardly-rectifying channel (Kir6.1) and Fluoro-Nissl dye [101,102]. PC coverage of the CNS microvessels is significantly higher (10–30-fold) than of the peripheral vessels [117], and PCs play an important role in regulating capillary diameter and cerebral blood flow [39,41,116,118,119], as well as ECM protein secretion and interactions [120]. Adjoining membranes of the neighbouring PCs are interconnected with GJs, serving as a functional syncytium along the microvascular wall [101,121]. Vasomotor response propagation consists of a local change in blood flow in response to local changes of neural activity that largely depend on GJ connections among PCs and ECs, but not to surrounding neurons and glia [122,123]. PCs strongly influence the expression of BBB-specific genes and proteins, and also polarize astrocyte end-feet surrounding CNS blood vessels, underlining their important role in both EC and astrocyte functional integration [5,124]. Animal models of CNS PC deficiency feature BBB impairment and microaneurysms [5,120], denoting their crucial role in BBB stability, permeability and in capillary blood flow regulation [125,126]. These models of CNS PC deficits also show EC hyperplasia, again highlighting the primary PC role in the regulation of EC proliferation, migration and stabilization during developmental and tumoral angiogenesis [7,99,127,128,129]. Another important role of forebrain PCs is to lead microvascular sprouting angiogenesis by raising tunnelling nanotubes (TNTs) and microtubes (MTs), tubular PC conduits surrounded by collagens, laminins, fibronectin [66,72,130,131]. PC TNTs and MTs are EC-free structures which primarily explore the surrounding tissue [132,133] and subsequently recruit ECs in the growing vasculature [19,72].

The heterogeneity and multitasking aptitude of CNS PCs have already been demonstrated in specific developmental and pathological conditions [107,134]. Different CNS PC subtypes have been described depending on their ontogeny [5,14,18,62,71,82,88,135]. Brain PCs have been classified in three subtypes: ensheathing, mesh and thin-strand PCs based on their morphology and location [18,134,136] (Figure 1).

PC protein expression also varies along the course of the microvasculature, presumably to accommodate differing functions [5,17,18,137,138]. PC transitional elements are observed along the vascular bed at various developmental stages or after pathological stimuli [5,6,137,139]. Additionally, the transition from arteriolar SMCs to pre-arteriolar PCs is not immediate; hybrid cells precede the ensheathing PCs and PCs with meshed, circular processes [6,18] (Figure 1).

By establishing tight relations with ECs, PCs contribute to human forebrain developmental angiogenesis and vessel branching [140,141], vascular network stability [10,110,111,127] and BBB functioning [118]. PCs are key regulators of EC functions, including EC proliferation, apoptosis with corresponding vessel regression [127], and BBB formation and integrity in (patho) physiological conditions [142,143]. PCs are required for the maintenance of metabolic neurovascular coupling to ensure an adequate blood supply in active brain regions, gliovascular control, as well as the promotion of neurogenesis within neurogenic niches or oligodendrogenesis within oligovascular niches [102,142,144,145]. A competent BBB provides an optimal microenvironment for neurons and glia, while a partially compromised BBB may support the high metabolic needs of neural stem cells and progenitors. In fact, a lower expression of tight junction (TJ) proteins and aquaporin-4 (AQP4) has been described in microvessels of the hippocampal subgranular zone and subventricular zone [146], where stem cells remain undifferentiated thanks to the absence of PC endfeet and to direct interactions with EC ephrinB2 and Jagged1 [147]. Differentiation of these progenitors is strongly supported by PC secretion of diffusible differentiating signals [148]. PCs also contact OPCs [69,145], and these two populations reciprocally regulate their reciprocal proliferation and survival [145,149,150]. The crosstalk between migrating OPC and vasculature starts early during development (at around 14 days in mouse embryonic brain and 14 weeks of gestation in human cerebral cortex) when parallel microvessels of cortical plate serve as scaffolds for OPC migration [19,151]. CNS PCs secrete paracrine factors [152] and support the highly demanding metabolism of myelinating oligodendrocytes [153]. PC-deficient mice show a disrupted white-matter microcirculation and BBB impairment with fibrinogen extravasation, which, in turn, are responsible for autophagy-dependent oligodendrocyte death [154]. PC dysfunctions are responsible for a disrupted EC metabolism and astrocyte endfoot localization and distribution [107,110,154].

Moreover, PCs are recognized as a key cellular component of BBB models in vitro such as the BBB-on-chip or brain-on-chip microphysiological systems, as well as being a promising tool for CNS regeneration [155,156,157,158,159,160]. In fact, in vitro BBB model studies revealed that PCs release high levels of Ang-1 and transforming growth factor beta-1 (TGFβ-1), acting on EC Tie-2 receptor and TGFβ/ALK1/endoglin receptors, respectively [107,134]. Decreased levels of the TJ molecule occludin are responsible for BBB dysfunction in disease [161]. In this context, a progressive reduction in occludin gene expression, as a result of a drop in PC-derived Ang-1 levels, may contribute to the loss of the BBB in several CNS diseases [134,136]. PCs are also required for the expression of EC major facilitator superfamily domain-containing protein 2a (MFSD2a), another important molecule for BBB formation and integrity [140,141]. Loss of PC coverage triggers breakdown of the BBB, increases EC transcytosis and interferes with the polarization of astrocytic endfeet at cerebral vessel wall surfaces [10,110,111,127]. An age-dependent reduction in TJ protein expression and consequent BBB dysfunction have been both observed in the aging human hippocampus [162,163] and in *Pdgfrβ* knockout mice [127]. Progressive dysfunction of PCs has been observed in different neurodegenerative diseases characterized by impaired O_2_ delivery to the cerebral tissue and a functional impairment of the NVU [164]. Indeed, the PC role in the molecular pathogenesis of NVU/BBB dysfunction is gradually gaining increasing attention in neurosciences and may provide interesting solutions to the urgent need for biomarkers of neurovascular unit (NVU) damage. A biomarker is a biological observation that can act as a substitute for a clinically relevant endpoint or intermediate outcome that is more difficult to observe [165]. In recent years, innovative methods for molecular or imaging biomarker detection have been implemented in neurological settings. Foremost among them, the detection of biomarkers such as NfL, Tau, or GFAP in the CSF and in lower concentrations in the blood has a significant diagnostic impact [166]. New technologies have enabled the quantification of nanomolar concentrations of brain-derived biomarkers in blood, allowing a minimally invasive diagnosis of brain damage and neurodegenerative processes [167]. The appearance of NVU molecules in blood has been reported for neurodegenerative diseases [168,169], seizures [170,171,172,173,174], neurologic manifestations of systemic disease [175], traumatic brain injury (TBI) [176,177,178], psychiatric diseases [179] and brain tumours [180,181].

## 7. PC Dysfunction in Neurodevelopmental Disorders

Relatively less attention has been directed to the importance of PC dysfunction in neurodevelopmental disorders than in neurodegenerative diseases. However, analysis of postmortem sections of autism spectrum disorder patients evidenced signs of angiogenesis in all brain areas analysed, with a significant increase in nestin^+^ PCs and CD34^+^ ECs in comparison with age-matched controls [182]. In addition, considering that a dysfunction of innate immune signalling pathways has been functionally linked to neurodevelopmental disorders, including autism and schizophrenia [183], the relationships between PCs and microglia during brain development should be assigned a capital importance. In fact, a model of specific PC depletion induced by the intraventricular administration of a neutralizing antibody against PDGFRβ elicited failure of the microglia to promote the differentiation of the neural stem cells, with possible indirect effects on neural circuitries and neurodevelopmental processes [184]. In addition, inflammatory cells that have passed through the leaky BBB, such as macrophages and cytokines (interleukin-1β, tumour necrosis factor-α) have been found in white matter lesions in postmortem studies of cerebral palsy [185]. An increased blood–brain barrier permeability to proteins from plasma was specifically induced within the microvessels of white matter tracts in postnatal rats and caused an increase in oxidative damage to OPCs and consequently poor myelination [186,187,188]. A reduction in PC coverage (α-SMA and desmin) of brain microvessels was observed in a model of chronic hypoxia during prenatal life [189,190].

## 8. PC Dysfunction in Alzheimer’s Disease (AD)

Dysfunction of the NVU is a well-recognized feature of Alzheimer’s disease (AD) [191,192,193,194,195,196,197]. In addition to the other pathogenic models of AD onset and development, based on amyloid deposition, mitochondrial calcium (Ca^2+^) signalling and glial dysfunctions [198,199,200], the so-called cerebral amyloid angiopathy (CAA) model, that consists of the accumulation of beta-amyloid (Aβ) in leptomeningeal and cortical feeding arteries of cerebral and cerebellar lobes, has also been studied [201,202]. A combination of many factors, including an initial chronic brain hypoperfusion, abnormal microvascular remodelling, BBB breakdown, progressive accumulation of ischemic lesions and microhaemorrhages associated with neuroinflammation and Aβ deposition, NVU disorganization, loss of synaptic plasticity and neuronal deaths, results in progressive cognitive and behavioural deficits [202,203,204]. All components of the NVU could be affected in CAA and AD: EC and SMC degeneration in medium-sized cerebral vessels and EC, PC, perivascular astroglia and ECM disorders in cerebral microvessels have been described along the time-course of AD and CAA progression [205,206,207,208]. However, microvascular PCs have appeared to play a significant role in the progression of AD-associated microvascular alterations and impairment of plasticity in AD brain [117,209], although PC loss should be considered with some caution, because the morphology and functions of PCs too could be compromised in the aged brain without dementia [210]. In the adult brain, PCs show a great plasticity, reducing in number but extending and retracting their processes to prevent vascular dilation and exposure of the abluminal side of brain microvascular ECs [210]. However, AD neurodegeneration is accompanied by significant, progressive PC dysfunctions and apoptosis with BBB breakdown and neuroinflammation [211,212] due to the toxicity of Aβ [213,214] and of reactive oxygen species [215,216], the pro-apoptotic action of advanced glycation end products (AGE) on their PC receptor (RAGE) [217] and to excessive PC remodelling due to angiogenic stimuli [218,219]. Dysfunctional microvascular PCs release soluble PDGFRβ through ADAM10 activity; PDGFRβ has been proposed as a biomarker of BBB integrity and the PDGFRβ increase predicts neurodegeneration [220,221,222]. Quantification of cerebrospinal fluid PDGFRβ has recently been developed using a new, highly sensitive method, which could be extended to study other neuropsychiatric diseases with NVU/BBB impairment [223].

In a mouse model of AD, the APPswe/PS1 mice fed with a high-fat diet, an accelerated, progressive dysfunction and apoptosis of PCs have been described [224,225]. On the other hand, functionally competent PCs contribute to maintaining an adequate blood supply to the brain and provide effective clearance of Aβ [117,226], to regulate insulin transport to the cortical neurons [227], and increase the insulin sensitivity of hypothalamic neurons [228], preserving the brain from dementia. Cerebrovascular PCs have also been implicated in the production of lipidated, reactive forms of apolipoprotein E (ApoE) acting together with perivascular astroglial cells, and, in turn, ApoE suppresses PC motility and adhesion to ECs [229]. In PCs, ApoE regulates the cyclophilin A (CypA)–NF-κB–matrix metalloproteinase (MMP-9) pathway, which is necessary for cerebrovascular integrity, because ApoE insufficiency leads to an elevated PC production of CypA and consequent BBB breakdown [230]. ApoE is a cholesterol carrier also implicated in the repair of brain injury. APOE polymorphic alleles are the main genetic determinants of the AD risk: the ε4 allele confers an increased risk of AD compared with the more common ε3 allele, whereas the ε2 allele decreases the risk. The APOE ε4 allele also confers an increased risk for CAA and age-related cognitive decline during normal ageing. Individuals carrying the ApoE ε4 genetic variant display severe alterations of the structural and functional integrity of the BBB, characterized by TJ disruption, PC loss in brain microvessels due to excessive activation of low density lipoprotein receptor-related protein 1 (LRP1)-dependent CypA–MMP-9 signalling induced by ApoE4 expression in PCs [194]. Another early neuropathological finding in the AD brains is the prominent oligodendropathy (Figure 2, Table 2) [231], which is associated with a PC inability to support oligodendrogenesis, as already observed in multiple sclerosis patients and models [232,233].

## 9. PC Dysfunction in Multiple Sclerosis (MS)

As already stated, PCs and OPCs reciprocally interact to ensure myelin development and maintenance and BBB integrity at the NVU interface [69,145,149,150,153,154]. An altered interaction leads to demyelinating events and disturbs remyelination attempts of demyelinating plaques in MS and animal models of demyelination/remyelination [69,232]. MS is a progressive autoimmune disease of unknown aetiology that affects the CNS of young adults, in which immune cells react against myelin antigens. It is characterized by early BBB breakdown preceding multiple demyelinating lesions which commonly include gliosis, axon degeneration, neuronal loss and remyelination attempts [234,235,236,237,238]. The evidence supporting a role of microvascular brain PCs in MS derives from detailed analyses of post-mortem brain tissues from MS-affected patients at an early phase of their secondary progressive MS course in comparison with others at a late phase of their progressive course. Perivascular cell subpopulations (including PCs) behave differently depending on the lesion type and clinical course of the disease [239]. Active lesions contain higher numbers of proliferative perivascular cells than inactive lesions, while chronic lesions display lower numbers of proliferative perivascular cells than normal-appearing white matter (i.e., healthy tissue external to demyelinating plaques).

Apart from neuropathology, another substantial contribution to the MS research field has been provided by animal models built in order to reveal cellular and molecular mechanisms of demyelination/remyelination and to develop novel approaches to MS treatment. Experimental autoimmune encephalomyelitis (EAE) in rodents is the animal model most commonly used to study demyelination/remyelination, neurodegeneration and neuroinflammation processes that closely resemble features of human MS [240]. Other experimental models specifically developed to study demyelination/remyelination processes are based on the administration of chemicals that promote systemic or focal CNS demyelination [241]. In EAE, NG2^+^ OPCs make many more contacts with NG2^+^ PCs of cerebrocortical microvessels, determining TJ impairment and BBB leakage (Figure 3, Table 2). The absence of NG2, in both OPCs and PCs, reduces the number of vessel-contacting OPCs and restores the BBB integrity [69,144,242,243].

In a model of focal demyelination of cerebellar peduncles of adult rats built using lysolecithin, perivascular PDGFRβ^+^ PCs proliferate in close proximity to differentiating OPCs [144]. In parallel, another proliferating population of PDGFRβ^+^ cells, denoted as parenchymal or CNS-PCs-like cells, develops within the lesion [144]. CNS PCs induce the differentiation of new oligodendrocytes from OPCs because of their secretion of the laminin alpha2-chain in the microvascular basal lamina [149]. This promoting effect of PCs on promoting oligodendrocyte differentiation has also been demonstrated in adult neural stem cells [149]. The discovery that the paracrine pro-oligodendrocyte effect of CNS PCs depends on the expression of the scaffolding protein A-kinase anchor protein 12 (AKAP12), which interacts with protein A-kinase/cyclic adenosine monophosphate response element-binding protein transcription factors, points in the same direction. PCs that lack AKAP12 expression lose their paracrine role in promoting OPC differentiation when compared with conditioned media harvested from wild-type CNS PCs [244]. Another important role played by PCs in MS pathology in their dynamic contribution to neuroinflammation concerns the expression of the purinergic receptor adenosine triphosphate (ATP)-gated P2X receptor cation channel (P2X7R), relevant for triggering the inflammatory response, by PDGFRβ^+^ PCs [245]. In EAE, the overexpression of P2X7R on the PC surface has been associated with a downregulation of PDGFRβ expressed by PCs, and of the TJ molecule claudin-5 in ECs, while administration of a P2X7R antagonist restores the expression of PC PDGFRβ, EC claudin-5, and the BBB competence and improves the clinical status of affected mice [245]. In addition, during EAE PCs exhibit a dynamic behaviour and heterogeneous morphology, and feature an increase in PC coverage length along the microvessel wall during the disease course, and a concomitant decrease in PC density. A notable elongation was observed in inflamed perivascular cuffs of EAE [246]. Another important PC role contributing to neuroinflammation in MS is their ability to secrete several adhesion molecules and chemokines/cytokines that assist in the recruitment and migration of monocytes, T cells, eosinophils and neutrophils [161,247,248,249,250]. PCs also express pro-inflammatory factors, such as interleukin-1β, tumour necrosis factor-α and vascular endothelial growth factor, that can induce a pro-inflammatory state in astrocytes, microglia and ECs, and recruit leukocytes [246,251,252]. Chondroitin sulfate proteoglycans (CSPGs), a family of ECM proteins comprising the transmembrane CSPG NG2, a marker of active PCs, are enriched within inflamed perivascular cuffs of MS lesions, where they inhibit OPC differentiation in oligodendrocytes [253]. CSPGs are believed to raise the levels of pro-inflammatory chemokines/cytokines in macrophages [254] and PC culture media [246]. Direct interactions have been demonstrated, since PCs stimulated with CSPGs enhanced macrophage migration [246]. On the other hand, PC viability is drastically reduced following incubation with sera derived from MOG-induced EAE-affected mice and from primary progressive and secondary progressive MS patients [250]. Increased angiogenesis has been observed in normal-seeming grey and white matter in MS [255,256,257] and in different animal models of MS [258,259,260,261,262,263,264,265].

All these findings, taken together, indicate that during MS, CNS PCs display phenotypic changes according to the pathological progression that involve inflammation, demyelination, eventual remyelination, axonal death and scar formation. Therefore, CNS PCs have been suggested to be an attractive therapeutic target for the treatment of MS [266].

## 10. PC Dysfunction in Stroke and NeuroCOVID-19

PCs are highly vulnerable to ischemic injury and appear to be responsible for an unfavourable impact on stroke-induced tissue damage and brain oedema by disrupting the microvascular blood flow and BBB integrity. As already pointed out, PCs contribute to match the metabolic demand of nervous tissue with the local microvascular blood flow (neurovascular coupling) through vasomotion [5,119,267,268,269], and also contribute to leukocyte trafficking across the BBB [246,270,271,272,273,274]. Ensheathing PCs, which are located at the level of penetrating arterioles and pre-capillaries, embrace ECs [269,275,276]. Transitional PCs, characterized by circumferential, merging processes with a rich in α-SMA scaffold (Figure 1), are observed along the vascular bed at various developmental stages or after pathological stimuli and show a transitional phenotype between SMCs and PCs [6,18,110,137,139,277]. PC subtypes and SMCs both respond to neural activity by the regulation of vascular diameters through α-SMA myosin-based contraction [278,279,280] activated by receptors for vasoactive mediators expressed on their surface [101,102,209,278,281,282,283,284,285,286,287]. Capillary dilation mediated by PC resting has been observed even before arteriole dilation in response to a very focal request derived from a small group of nearby neural cells, as a final step of flow regulation, while the arterioles control a larger portion of neural tissue [102]. However, not all microvascular PCs are able to contract, and the proportion of contractile ones may vary depending on the tissue, species, developmental stage and localization along the arteriovenous length [285,288,289]. Neuromicrovascular coupling is mediated by astrocytic Ca^2+^ entry through ATP-gated channels, activating the release of vasodilatatory molecules derived from arachidonic acid via phospholipase D2 and diacylglycerol lipase rather than phospholipase A2. In contrast, the dilation of arterioles depends on N-methyl-D-aspartate receptor activation and Ca^2+^ dependent nitric oxide generation by interneurons. Vasodilation was shown to occur via the formation of prostaglandin E_2_ (PGE_2_) and epoxyeicosatrienoic acids, while production of 20-hydroxyeicosatetraenoic acid resulted in vasoconstriction [290]. Glutamate release generates PGE_2_, which dilates the capillaries by activating an outward potassium (K)^+^ current in PCs [102]. Similarly, PGE_2_ activates an outward K^+^ current in aortic smooth muscle [291] and relaxes kidney PCs [292]. Transgenic mice with a decreased number of PCs had a deficient neurovascular coupling [209]. A recent report has demonstrated that brain PCs have distinct signatures of Ca^2+^ activity depending upon their localization within the vascular tree and neuronal activity [293]. Ca^2+^ signalling is essential in modulating the PC contraction [269]. Ensheathing PCs have been shown to exhibit highly regular oscillatory Ca^2+^ fluctuations like those of SMCs and to possess an important contractile potential to regulate the microvascular blood flow, while capillary and venular PCs, which are less efficient in regulating microvascular blood flow, exhibited irregular Ca^2+^ signals of irregular frequencies [293]. Interestingly, a deregulation of this contractile PC function is directly implicated in the pathogenesis of cerebrovascular disorders [5,294,295]. During ischemia, it was shown that in situ PCs constricted capillaries through a Ca^2+^-induced α-SMA contraction and subsequently died, impairing the microcirculatory re-flow after recanalization [101,102,296]. Energy loss during cerebral ischemia leads to an uncontrolled overload of intracellular Ca^2+^ in the PCs of the ischemic core and penumbra [121]. This Ca^2+^ overload is probably potentiated by reactive oxygen species (ROS) originating from multiple sources during ischemia-reperfusion [297,298] including PC mitochondria, astrocyte endfeet and ECs [277,299,300]. PCs express one of the major superoxide-producing enzymes, nicotinamide adenine dinucleotide phosphate oxidase 4 [301,302,303]. In a stroke model, this enzyme is upregulated in microvascular PCs of the peri-infarct region and contributed to the activation of MMP-9 and BBB leakage [303,304]. ROS and peroxynitrite are abundantly generated during ischemia and reperfusion and further contribute to a prolonged PC contraction [296,300,305]. This prolonged contraction impairs tissue reperfusion despite recanalization, and is known as the ‘no-reflow phenomenon’ [116,306,307,308,309]. Microvessel lumina at the constricted segments are filled with entrapped erythrocytes, leukocytes and fibrin-platelet deposits [310,311]. Anti-thrombotic agents and genetic manipulations reducing microvascular clogging by inhibiting leukocyte adherence, platelet activation, or fibrin-platelet interactions have been shown to restore the microcirculation and improve the stroke outcome in animal models [312,313,314,315,316]. Current guidelines, however, do not recommend anti-thrombotic medication use during concomitant recanalization therapies because of the increased risk of haemorrhage [317,318,319]. Interestingly, adenosine-squalene nanoparticles have been shown to improve the microcirculation by relaxing contracted PCs during ischemia in mouse stroke models [320]. Using simultaneous imaging of ROS formation in the parenchyma and vasculature, 2-sulfo-phenyl-N-tert-butyl nitrone, a BBB-impermeable ROS scavenger, also provided neuroprotection by improving microcirculatory reperfusion with a not easily comprehensible relaxing effect and then secondarily reducing parenchymal ROS formation without entering the parenchyma [321]. Consequently, the restoration of the microcirculatory reperfusion emerges as an exciting target to improve the success rate of cerebro-afferent vessel recanalization and neuroprotection therapies [299,322,323]. The presence of microcirculatory damage distal to the thrombus prior to attempting recanalization is an unfavourable prognostic factor for the clinical outcome in acute ischemic stroke patients treated with clot retrievers [324]. Injury to PCs during acute ischemia contributes to BBB breakdown through the release of MMP-9, hence brain oedema in the ischemic territory, in addition to impairing tissue microcirculation [325,326]. In this context, early detection of biomarkers of NVU damage can improve the therapeutic window for time-sensitive brain damages such as stroke and TBI [327,328,329]. The availability of salivary point-of-injury sampling for neuron-specific enolase, S100B and microRNAs offers an opportunity for economical, simple and safe exploitation of BBB permeability [177,330,331,332,333,334]. Considering the increased PDGF signalling during cerebral ischemia [150,335], plasma levels of PDGF-AA/ BB/AB could be used as early biomarkers of stroke duration and outcome prediction [336,337,338]. This PC-attracting signal mobilizes PCs from microvessels of the peri-infarct areas within 1 h of ischemia. This PC migration may be protective because it provides guidance for peri-infarct angiogenesis and neurogenesis thanks to the ability of PCs to secrete growth factors in response to ischemia-triggered signals [130,339,340,341]. Conversely, PC activation can be detrimental as they could increase microvascular permeability by disrupting the interaction with EC TJs [342,343] and subpopulations of PDGFRβ^+^ activated PCs contribute to the fibrotic reaction upon brain injuries by an upregulation of α-SMA and release of ECM molecules in infarct area [344]. In the longer term period, however, post-stroke angiogenesis and neurogenesis in the peri-infarct area play important roles in stroke outcomes [345,346].

Other important stroke risk factors are diabetes mellitus and arterial hypertension. During diabetes, the loss of PC coverage around retinal ECs has been shown to trigger pathological angiogenesis, EC apoptosis and plasma leakage (Figure 4, Table 2) [347,348,349,350]. Although the effects of diabetes on brain PCs have not been fully elucidated, a decreased PC density has been reported within the cerebral microcirculation [351]. Experimental strokes in diabetic mice have led to an increased occurrence of haemorrhagic transformation after ischemia [352] and the impairment of vascular repair mechanisms critical for neovascularization and angiogenesis [347]. The same phenomenon has been observed in patients with acute ischemic stroke [353]. In hypertensive animals, cerebral PCs show irregular profiles, associated with fragmented processes and thickening of their basement membranes [354]. These changes reportedly led to a decreased endothelial coverage by PCs, capillary thrombotic occlusion and luminal collapse [354]. Capillary dysfunction induced by the above cerebrovascular disease risk factors has also been proposed to contribute to the risk of a subsequent stroke and cognitive decline [355].

In addition, brain PCs abundantly express angiotensin-converting enzyme 2 (ACE2) at the neurovascular interface. Their response to severe acute respiratory syndrome coronavirus-2 (SARS-CoV-2) infection is starting to be elucidated and could be important in understanding microvascular injury in coronavirus disease 19 (COVID-19) [356,357]. SARS-CoV-2 causes COVID-19 [358], a variable respiratory illness with frequent vascular-mediated neurological complications [359,360]. COVID-19 pathogenesis mechanisms are elusive. SARS-CoV-2 infects host cells via the binding of viral spike protein to the transmembrane receptor ACE2 [361]. ACE2 is restricted to a subset of neurovascular PCs and its expression level is correlated with neurological symptoms [357]. ACE2 was also expressed to a much lower extent on ECs, macrophages and glia at the NVU interface [356,362]. Currently, it is hypothesized that there is a direct passage of SARS-CoV-2 through a dysfunctional BBB to actively infect ACE2^+^ PCs [356,363]. Multifocal thrombotic microangiopathy and a PC pro-coagulant state were reported in the brains of patients who died immediately after the resolution of a SARS-CoV-2 infection [364,365,366,367]. SARS-CoV-2 dsRNA was identified in the vascular wall infiltrated by T cells and macrophages [357]. In addition, global or focal hypoxic/ischemic zones associated with large or small infarcts, and accompanied by microglial activation, were reported in the brain of COVID-19 patients [364,366,367]. Exposure to the SARS-CoV-2 spike protein causes the transition towards a vasoconstrictive PC phenotype characterized by an increased frequency and synchronicity of intracellular Ca^2+^ in PCs and the acquisition of an ensheathing PC Ca^2+^ waves and morphology, both caused by Notch3 signalling impairment [368]. These PC dysfunctions are in line with a recent paper indicating that SARS-CoV-2 binding to ACE2 triggers PC-mediated angiotensin-evoked cerebral capillary constriction [369]. In addition, the SARS-CoV-2 spike protein also deregulates the immune functions of brain PCs through the activation of the nuclear factor kappa-light-chain-enhancer of activated B cells (NF-κB) signalling pathway [368].

## 11. Conclusions and Perspective

Many cerebrovascular and neurodegenerative diseases are characterized by concomitant hypoxia of the cerebral tissue and functional impairment of the NVU [164]. PCs have a central role in preserving the brain microenvironment and supporting NVU function because their function maintains BBB physiology, due to the complex interdependency among the different components of the NVU. PCs are important for BBB differentiation during embryonic development and for maintaining a healthy and robust BBB in adulthood. A perturbed BBB exacerbates the progression of injury in cerebrovascular and neurodegenerative diseases. Nevertheless, our understanding of BBB regulation during cerebrovascular and neurodegenerative diseases remains rudimentary, thus permitting only slow progress in the fight against these neurological diseases. A better understanding of perivascular cell signalling during cerebrovascular and neurodegenerative diseases would provide a better insight. NVU adaptation to injury alters their signalling profiles and secretion of various injury-regulated factors [5,379], and thereby modifies cell–cell interactions and crosstalk [31,101,170]. Identifying key drivers of such responses, and discovering how to prevent PC dysfunction and how these cells influence the outcome, could thus provide new ways to modulate BBB function, to modify the outcome of recanalization therapies and prevent haemorrhage and oedema [363]. Hopefully, in the coming years we will learn more about the role of PCs in CNS diseases, and the use of PC biomarkers to measure NVU dysfunctions may allow us to devise approaches to more effective treatment.

## Figures and Tables

**Figure 1 cells-11-01707-f001:**
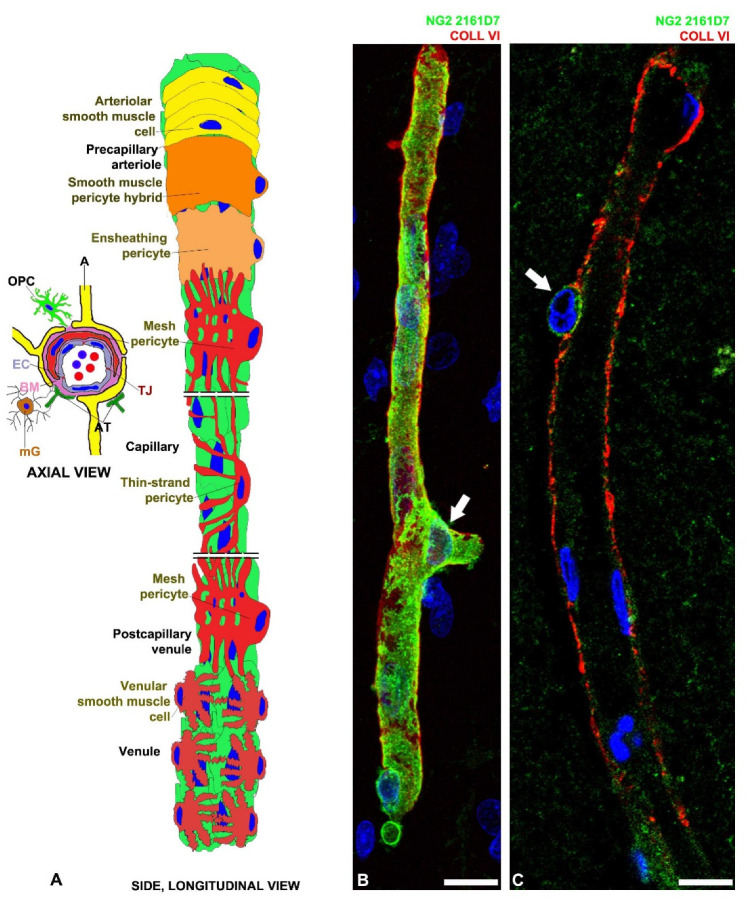
(**A**) A schematic depiction of NVU components from axial and longitudinal views: endothelial cell (EC), pericyte (PC), perivascular basement membrane (BM), and astrocytes (A), vessel-associated microglial cells (mG), OPCs/NG2-glia, macrophages, axon terminals (AT) which can contact EC. The longitudinal drawing shows mural cell types along the CNS vasculature without considering the presence of continuous BM between ECs and PCs. Smooth muscle cells wrap the arterioles, forming multiple concentric rings in continuity with hybrid smooth-muscle-PCs residing along the precapillary arterioles, which appear to join to ensheathing PCs at the arteriole–capillary interface. PCs in microvessels typically exhibit two phenotypes: the mesh PC characterized by short ramified processes, versus the thin-strand PC with long processes embracing the microvessel in a single strand, or helically twisted strand pairs. Mesh PCs prevail at the capillary–postcapillary venule interface. Stellate-shaped smooth muscle cells are present around parenchymal venules. (**B**) The mesh PC fine morphology and microvascular basal lamina relationships are shown in a representative vessel of the dorsal wall of the telencephalic vesicles (forebrain, future neocortex) of a 22-week-old human foetus. The extensive PC coverage and its relation to the collagen VI-enriched basal lamina is highlighted by an NG2/CSPG4 isoform, specifically recognized by the antibody 2161D7, that outlines not only the finer cell details such as the dense net of finger-like processes, but also the abluminal bumpy surface of the PC body (arrow) at the branching point. (**C**) The same 2161D7 antibody, recognizing an NG2/CSPG4 isoform expressed by foetal brain PCs, weakly stains just the PC body (arrow) of the adult human parahippocampal cortex. Other monoclonal antibodies against NG2 isoforms and commercial antibodies do not stain human adult CNS PCs [19]. Nuclear counterstaining TO-PRO3. Bars B, C: 10 µm.

**Figure 2 cells-11-01707-f002:**
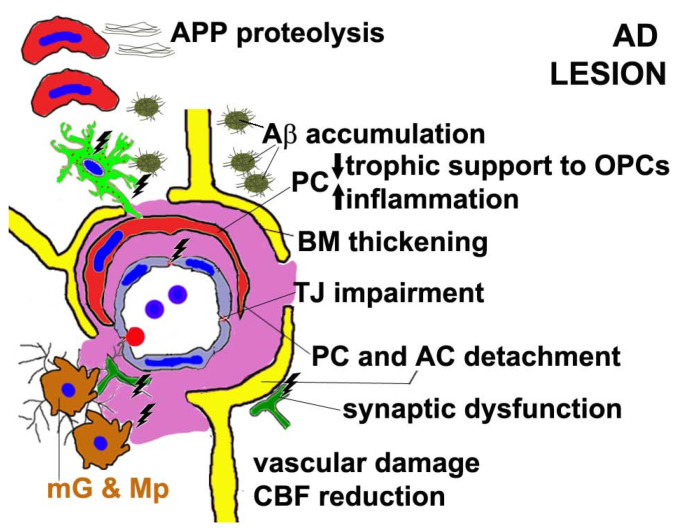
A schematic depiction of NVU components in Alzheimer’s disease (AD). The pericyte (PC) and astrocyte (AC) endfeet both appear detached from the vessel lumen, the perivascular basement membrane (BM) is thickened and vessel-associated microglial cells (mG) and macrophages (Mp) induce neuroinflammation. Dysfunctional PCs also release pro-inflammatory molecules and reduce the trophic support to oligodendrocyte precursor cells (OPCs)/NG2-glia. Altogether, these dysfunctions induce vascular damage and a reduced cerebral blood flow (CBF), acting via altered amyloid precursor protein (APP) proteolysis and consequent amyloid β accumulation inducing synaptic dysfunction and neuronal loss.

**Figure 3 cells-11-01707-f003:**
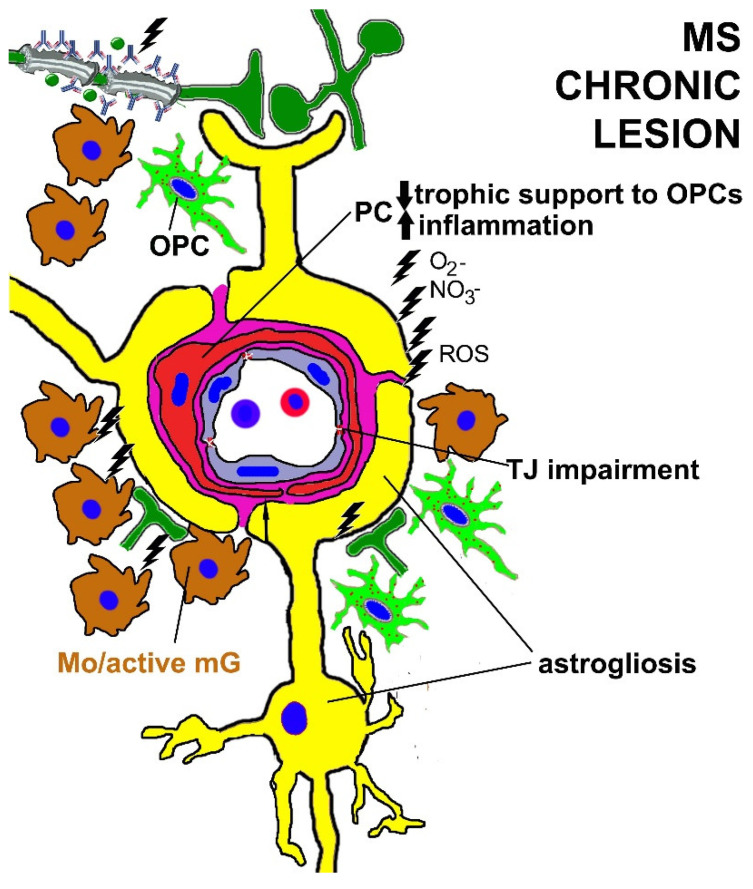
A schematic depiction of NVU components in a chronic end-stage MS lesion dominated by hypoperfusion and a persistent inflammatory milieu with abundant reactive oxygen species (ROS, O_2−_), peroxynitrite (NO_3−_). Vessel-associated microglial cells, monocytes, macrophages (Mp) and autoantibodies induce persistent neuroinflammation, also responsible for pericyte (PC) dysfunctions. PCs release pro-inflammatory molecules and reduce the trophic support to oligodendrocyte precursor cells (OPCs)/NG2-glia. The drawing also shows the inflammatory influence of demyelination on reduced axonal activities and vasoconstriction. Hypoperfusion is also attributable to vessel wall hyalinization, collagen deposition and astrocyte endfeet hypertrophy.

**Figure 4 cells-11-01707-f004:**
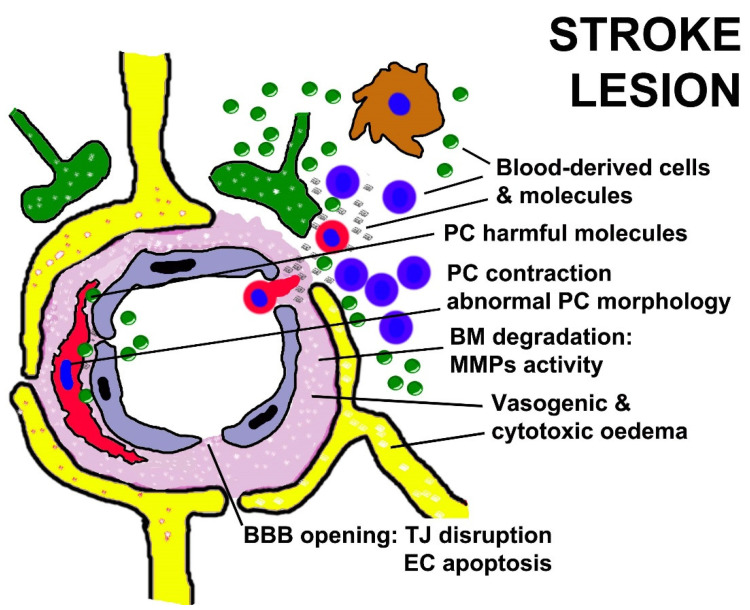
A schematic depiction of NVU components in a stroke lesion dominated by BBB leakiness, vasogenic swelling of BM, cytotoxic oedema, a PC hypercontracted state, astrocyte endfeet detachment from PCs, and completely disrupted NVU interactions: the TJ proteins claudin-5 and occludin are damaged and EC apoptosis is frequently observed; the BM is degraded by MMPs, mainly released by leukocytes infiltrating vessel BM and PCs enwrapping altered endothelial cells; large amounts of pro-inflammatory mediators and fibrin(ogen) derive from the blood stream, but additional harmful inflammatory molecules are released by PCs.

**Table 1 cells-11-01707-t001:** Best known markers of CNS PCs and their dynamic expression during development and adulthood.

PC Marker	Development	Healthy Adulthood
Neuron-glial antigen 2 (NG2)	[19,63]	Only activated PCs of CNS [64,65]
Platelet derived growth factor receptor beta (PDGFRβ)	[66,67]	[68]
Alanyl aminopeptidase (CD13)	[69]	[70,71]
Melanoma Cell Adhesion Molecule (CD146)	[71,72]	[73]
Endosialin (CD248)	[66,74]	Not expressed
Fluoro-Nissl dye NeuroTrace 500/525	[75]	[76]

**Table 2 cells-11-01707-t002:** Summary of PC dysfunction during clinical and experimental CNS conditions.

Disease	PC Dysfunction(s) in the Clinical Condition	PC Dysfunction(s) in Models
Autism spectrum disorder	Increased density [182]; genetic risk after PC gene modifications [370]	Putative NVU dysfunction [371]
Prenatal life hypoxia		Reduced PC coverage [189,190]; increased angiogenesis and bFGF [372]
White matter degeneration	[373]	Reduced PC coverage [154]
Altered neuronal circuitry	[120]	Ablation of PC induce poor interaction between glia and neural precursors [184,374].
AD	Hypoperfusion, PC degeneration and BBB breakdown [117,162,193,194,206,207,209,375]; PC inability to support oligodendrogenesis [233]	Decreased density of PDGFRβ^+^ PCs in 5xFAD mice [211] and CD13^+^ PCs in pericyte-deficient APP^sw/0^; Pdgfrβ^+/−^ [213]; dysfunction and apoptosis of desmin^+^ PCs in APPswe/PS1 mice [225]
MS	PC inability to support oligodendrogenesis [232]; increased density of PDGFRβ^+^ CD146^+^ PC in active MS lesions [239]; increased angiogenesis [255,256,257]	PC inability to support oligodendrogenesis in different models [69,144,145,149,150,154,232,244,253]; PDGFRβ^+^ PC triggers inflammation [245,246,247,248,249,250,251,376,377,378]; increased angiogenesis [258,259,261,262,263,264,265]
Stroke	Microvascular PC contractions aggravate ischemia [121]; PCs release harmful molecules [297]	In different models, PCs release harmful molecules [184,225,248,254,352]; decreased PC density, EC apoptosis, and BBB disruption [347,348,350,351]; abnormal PC morphology [354]
NeuroCOVID-19	SARS-CoV-2 infects ACE2^+^ PCs [356,363,365]	PC vasoconstriction [368,369]

## Data Availability

Not applicable.

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
