# Peer review of "Central Nervous System Pericytes Contribute to Health and Disease"

_cells, 2022, doi:10.3390/cells11101707_

Round 1

Reviewer 1 Report

The authors have written a comprehensive review on pericytes which includes sections on classifications of the cells, origin, molecular markers, functions, and dysfunctions in CNS diseases.

The content discussed in the manuscript is very relevant for researchers working on understanding the BBB function and modulation by components of the NVU of which pericytes are one of the components. The highlights on pericytes dysfunctions in specific CNS diseases are indeed useful to direct researchers toward exploring new avenues in CNS disease therapy.

This review will be of interest to the scientific community and I recommend acceptance of the manuscript for publication in Cells.

Author Response

We would like to thank the editor and the referee for their kind comments, that have allowed us the chance to improve this manuscript.

Reviewer 2 Report

The review by Girolamo et al titled Central nervous system pericytes contribute to health and disease is a well written comprehensive review that discusses in details the various functions of pericytes in healthy and disease state.

There are few comments that would strengthen this work:

I would suggest that the authors enhance the quality of Figure 1 as the colors are obscuring the labels of the figures.

Also, I would suggest that they would design a new schematic of the neurovascular unit showing the BM, ECM, Glia, EC and PCs that would highly benefit the readers to appreciate the NVU structural role and contribution to the BBB.

In addition, a small table that summarizes the different markers of PCs and their dynamic expression would be informative.

As forth sections discussing the disease and PCs, \I would recommend that the authors include tables to delineate the different expression or changes of PCs or their markers in theses disease condition both experimental and  clinical ones.

Finally, there have been high interest of brain injury and PC as contributors to the BBB breaching, the authors can add brain injury to their list of diseases.

Author Response

Reply

We would like to thank the editor and the referee for their kind comments, which have allowed us the chance to improve this manuscript. All the revisions to the manuscript are evidenced.

English language and style are fine/minor spell check required
I would suggest that the authors enhance the quality of Figure 1 as the colors are obscuring the labels of the figures.

Reply: We agree with the reviewer's suggestion and have modified the different colors of pericyte types to a darker brown
Also, I would suggest that they would design a new schematic of the neurovascular unit showing the BM, ECM, Glia, EC and PCs that would highly benefit the readers to appreciate the NVU structural role and contribution to the BBB.

Reply: We thank the reviewer for this suggestion and have modified the axial view of a microvessel in Figure 1.
In addition, a small table that summarizes the different markers of PCs and their dynamic expression would be informative.

Reply: We have added a small Table to the PC marker paragraph.
As forth sections discussing the disease and PCs, \I would recommend that the authors include tables to delineate the different expression or changes of PCs or their markers in theses disease condition both experimental and  clinical ones.

Reply: We have added a summarizing Table at the end of the manuscript.
Finally, there have been high interest of brain injury and PC as contributors to the BBB breaching, the authors can add brain injury to their list of diseases.

Reply: We thank the reviewer for this suggestion, but the PC contribution to brain injury is a so relevant and complex a field that it would need to be investigated in a future work.

Reviewer 3 Report

The role of pericytes is increasingly getting evident in Alzheimer’s diseases, multiple sclerosis, and other related diseases. This review describes the contribution of pericytes in neurovascular diseases and dementia.

Here are some comments which authors may wish to address.

  1. A table listing the diseases involved with pericytes and their consequences, and characteristics can be included. Also reference to previous work of animal and humans should be included in this table. This will provide an overview of the review to the readers
  2. It would be nice to include some schematics for some sections of the review such as Alzheimer’s disease, multiple sclerosis, and stroke demonstrating possible involvement of pericytes.

Author Response

We would like to thank the editor and the referee for their kind comments, that have allowed us the chance to improve this manuscript. All the revisions to the manuscript are evidenced.

The role of pericytes is increasingly getting evident in Alzheimer’s diseases, multiple sclerosis, and other related diseases. This review describes the contribution of pericytes in neurovascular diseases and dementia.

Here are some comments which authors may wish to address.

  1. A table listing the diseases involved with pericytes and their consequences, and characteristics can be included. Also reference to previous work of animal and humans should be included in this table. This will provide an overview of the review to the readers

Reply: We thank the reviewer for this suggestion and have added a summarizing Table at the end of the manuscript.

  1. It would be nice to include some schematics for some sections of the review such as Alzheimer’s disease, multiple sclerosis, and stroke demonstrating possible involvement of pericytes.

Reply: We have added 3 additional schematic depictions in the indicated sections.